# Effects of Isolated LAB on Chemical Composition, Fermentation Quality and Bacterial Community of *Stipa grandis* Silage

**DOI:** 10.3390/microorganisms10122463

**Published:** 2022-12-13

**Authors:** Mingjian Liu, Yu Wang, Zhijun Wang, Jian Bao, Muqier Zhao, Gentu Ge, Yushan Jia, Shuai Du

**Affiliations:** 1Key Laboratory of Forage Cultivation, Processing and High Efficient Utilization, Ministry of Agriculture, China, Key Laboratory of Grassland Resources, Ministry of Education, China, College of Grassland, Resources and Environment, Inner Mongolia Agricultural University, Hohhot 010019, China; 2National Engineering Laboratory of Biological Feed Safety and Pollution Prevention and Control, Key Laboratory of Molecular Nutrition, Ministry of Education, Key Laboratory of Animal Nutrition and Feed, Ministry of Agriculture and Rural Affairs, Key Laboratory of Animal Nutrition and Feed Science of Zhejiang Province, Institute of Feed Science, Zhejiang University, Hangzhou 310058, China

**Keywords:** *Stipa grandis*, isolation, lactic acid bacteria, silage quality, bacterial community

## Abstract

This study aimed to screen and identify lactic acid bacteria (LAB) strains from the *Stipa grandis* and naturally fermented silage, and assess their effects on the silage quality and bacterial community of *Stipa grandis* after 60 days of the fermentation process. A total of 38 LAB were isolated, and strains ZX301 and YX34 were identified as *Lactiplantibacillus plantarum* and *Pediococcus pentosaceus* using 16S rRNA sequences; they can normally grow at 10−30 °C, with a tolerance of pH and NaCl from 3.5 to 8.0 and 3 to 6.5%, respectively. Subsequently, the two isolated LAB and one commercial additive (*Lactiplantibacillus plantarum*) were added to *Stipa grandis* for ensiling for 60 days and recorded as the ZX301, YX34, and P treatments. The addition of LAB was added at 1 × 10^5^ colony-forming unit/g of fresh weight, and the same amount of distilled water was sprayed to serve as a control treatment (CK). Compared to the CK treatment, the ZX301 and YX34 treatments exhibited a positive effect on pH reduction. The water-soluble carbohydrate content was significantly (*p* < 0.05) increased in ZX301, YX34, and P treatments than in CK treatment. At the genus level, the bacterial community in *Stipa grandis* silage involves a shift from *Pantoea* to *Lactiplantibacillus*. Compared to the CK treatment, the ZX301, YX34, and P treatments significantly (*p* < 0.05) increase the abundance of *Pediococcus* and *Lactiplantibacillus*, respectively. Consequently, the results indicated that the addition of LAB reconstructed microbiota and influenced silage quality. The strain ZX301 could improve the ensiling performance in *Stipa grandis* silage.

## 1. Introduction

Typical steppe is mainly distributed in arid and semi-arid areas, especially in Inner Mongolia, China, which serves as an essential component of the terrestrial ecosystem [1], and plays a crucial role in promoting human well-being and providing sufficient nutrition to meet the demands of animal productions [2]. Hay is a traditional management practice in the utilization of natural grassland [3]. Generally, native grass hay is sufficient for animal feed during summer, but the supply can be limited in the autumn and winter when the forage grows slowly; therefore, it is hard to shake off an imbalance between the supply and demand of the ruminants [4]. In addition, the quality and yield of forage were mainly affected by the dominant species in natural grassland [5]. *Stipa grandis* was the dominant species of the typical steppe, and the application of which was limited due to the hard and sharp awn needles on the caryopses after maturity [6]. Therefore, it is necessary to look for a suitable technique for *Stipa grandis* to cover these shortages and promote its application in animal production.

Ensiling is a more practical way for forage preservation because of its fewer nutrient losses and good palatability [7]. However, the absence of lactic acid bacteria (LAB) counts, water-soluble carbohydrate (WSC) content, and moisture, associated with the high buffering capacity, are factors that often hamper natural grasses ensilage [8]. The inoculation of LAB has been proven to be a promising strategy to overcome this problem of poor fermentation [9], which can rapidly reduce the pH value and inhibit the activity of undesirable microorganisms for improving the ensiling performance of silage [10]. Zi et al. [11] reported that the addition of LAB can increase the abundance of desirable members of the bacterial community and eliminate harmful bacteria. Nevertheless, the reconstitution mechanism of LAB during ensiling is complex [12]. Fijałkowska et al. [13] proved that only suitable LAB strains can be used to guarantee successful silage. A previous study has reported that the best LAB strains for promoting fermentation quality always source from the forage itself [14]. LAB treatment has been isolated from multiple materials, including pineapple residue [15], defective banana [16], elephant silage [17], and alfalfa [18]. However, little knowledge is available on the LAB isolation and application from the dominant species of native grass, and to the best of our knowledge, there is no report on the *Stipa grandis* in LAB isolation and application.

Therefore, the purpose of this study was to characterize the isolated LAB from the dominant species of native grass using phenotypic, chemotaxonomic characteristics, and molecular methods and evaluate their effects on the silage quality and bacterial community of *Stipa grandis* silage.

## 2. Materials and Methods

### 2.1. Samples and Bacterial Isolates

The *Stipa grandis* for strain screening experiment was from a typical steppe in Maodeng Ranch, Xilinguole League, Inner Mongolia Autonomous Region, China, in August 2020, and was harvested at the booting stage. A total of 38 LAB strains from *Stipa grandis* silage and the fresh sample before ensiling were isolated using the methods of Cai et al. [19]. In detail, samples (10 g) of the fresh and fermented material were aseptically mixed with 90 mL of sterile distilled water and diluted to 10^−5^ based on 10-fold gradient dilutions. Serial dilutions from different gradients were spread to the de Man, Rugose, Sharpe (MRS) agar (Difco Laboratories, Detroit, MI, USA), and the plates were incubated at 35 °C for two days under anaerobic conditions. Thereafter, 2−3 colonies on MRS agar medium were picked randomly from each sample (*n* = 3) and purified third by streaking on MRS agar plates, and then stored at −80 °C in MRS containing 20% glycerol for further identification and analysis.

### 2.2. Physiological and Morphological Tests

Gram staining, catalase activity, and gas production from glucose were examined using the method given by Kozaki et al. [20]. Growth at different temperatures of the strains was detected in the MRS broth according to the procedure provided by Pang et al. [21]. Salt tolerance was tested in the MRS broth with NaCl at 3.0 and 6.5% for two days using the method of Li et al. [22]. Growth at different pH values was observed in the MRS broth (adjusting pH with 0.5 N HCl or NaOH) after incubation at 37 °C according to the method described by Liu et al. [23]. The API 50 CH (BioMerieux, Marcy l’Etoile, France) contains 49 different compounds and one control was selected to carry out the carbohydrate assimilation according to the instructions given by the manufacturer [24].

### 2.3. Identification of LAB Strains by 16S rRNA Sequence Analysis

The genomic DNA of screened strains was extracted by a TIANamp Bacteria DNA Kit (Tiangen Biotech Co., Ltd., Beijing, China) according to the instructions provided by the manufacturer. The genomic DNA concentration of each strain was measured by determining the ultraviolet absorbance at 260 nm with a spectrophotometer [25]. The 16S rRNA gene region of screened strains from different forages was amplified using the primers 27F (5′-AGAGTTTGATCCTGGCTCAG-3′) and 1492R (5′-TACGGCTACCTTGTTACGACT-3′) as the previous report [12]. The species identification was conducted according to previously described methods (Pitiwittayakul et al., 2021 [25]). To identify organisms, the 16S rRNA sequences were compared and identified with 16S rRNA sequences from Gene Bank by BLAST analysis [26].

### 2.4. Preparation of the Experimental Silages

*Stipa grandis* for the LAB addition experiment was harvested at the booting stage on 12 August 2021 in Xilinguole League, Inner Mongolia Autonomous Region, China. The material was not wilted after harvest and was shredded into lengths of 1–3 cm immediately with a manual forage chopper to determine its macronutrient composition. Two screened LAB strains, ZX301 with the accession number OP703379 and YX34 with the accession number OP703378, and a commercial LAB additive (strain number JYLP-326, defined as P, CH, Lactobacillus plantarum, Shandong Zhongke Jiayi Biological Engineering Co., Ltd., Shandong, China) was applied as silage additives. The LAB inoculation was designed as follows: *Stipa grandis* was inoculated with ZX301, YX34, and the commercial LAB strain at 10^5^ cfu/g fresh materials (FM), respectively, and the same amount of distilled water was sprayed to serve as a control [3,11]. Thereafter, approximately 150 g of silage material was tightly filled into polyethylene bags (22 cm × 32 cm; Shijiazhuang Youlang Trading Co., Ltd., Hebei, China) after mixing the cut plant material and additives thoroughly, and then sealed with a vacuum sealer to withdraw the air. Three replicates were prepared per treatment, and they were stored at room temperature (25 °C) for 60 days.

### 2.5. Chemical and Microbial Analyses

The dry matter (DM) content of fresh *Stipa grandis* and the samples after silage were calculated after oven-drying at 65 °C for 48 h until constant weight, and then pulverized to pass through a 1 mm screen using a sample mill for further analysis of chemical composition. The crude protein (CP) was calculated by multiplying 6.25 with the total nitrogen (TN) concentration, which was determined by the method of Kjeldahl [27]. The anthrone colorimetry was selected to evaluate the water-soluble carbohydrate (WSC) [28]. Neutral detergent fibers (NDF) and acid detergent fibers (ADF) were determined using an ANKOM A200i fiber analyzer (ANKOM Technology) according to the description of Van Soest et al. [29].

To determine the fermentation characteristics of *Stipa grandis* silage. Samples (10 g) were shaken well with 90 mL sterile aqueous solution by a homogenous slap apparatus to extract the fermentation broth, and then the four layers of gauze were used to filter the solution. The *Stipa grandis* silage extract was used to detect pH using a calibrated glass electrode pH meter (STARTED 100/B, OHAUS, Shanghai, China). The concentrations of lactic acid (LA), acetic acid (AA), propionic acid (PA), and butyric acid (BA) were quantified using a liquid chromatograph, according to the methods of You et al. [30]. The ammonia nitrogen (NH_3_-N) concentration was measured using the phenol-hypochlorite method according to Broderick et al. [31].

For microbial population analysis, the remaining subsample of approximately 10 g of fresh *Stipa grandis* and silages were 10^−1^ to 10^−5^ serially diluted with sterilized water, then spread on agar plates for microbial counting. Similar to the method of Guo et al. [32], the LAB was cultured and estimated by de Man Rogosa Sharpe agar (MRS) under anaerobic conditions, aerobic bacteria were counted by nutrient agar, coliform bacteria were enumerated on Violet Red Bile agar medium, and yeasts and molds were estimated by Potato Dextrose Agar. The number of colony-forming units (cfu) was expressed as a log on a fresh material (FM) basis.

### 2.6. Bacterial Community Sequencing Analysis

Total DNA from *Stipa grandis* and silage samples were extracted with a commercial sample DNA kit of E.Z.N.A. R (Omega Bio-tek, Norcross, GA, USA). The agarose gel electrophoresis (1%) and NanoDrop 2000 UV-vis Spectrophotometer (Thermo Scientific, Wilmington, United States) was used to measure the concentration and purity of extracted DNA according to the description of Huang et al. [33]. Primers targeting the V3−V4 regions of 16S rDNA (338F: 5′-ACTCCTACGGGAGGCAGCAG-3′; 806R: 5′-GGACTACHVGGGTWTCTAAT-3′) were selected to conduct PCR amplification, and which were performed by Biomarker Technologies Corporation (Beijing, China). Sequencing data for 16S rRNA gene sequence were stored in NCBI with BioProject accession number PRJNA891459.

### 2.7. Bioinformatics Analysis

Purified DNA was performed by Paired-end sequencing and the Illumina MiSeq PE300 platform (Biomarker Biotechnology Co. Ltd., Beijing, China). Thereafter, the low-quality reads were screened and trimmed to obtain high-quality clean reads according to the QIIME quality control process (version 1.9.1), then sequences were assembled using FLASH (version 1.2.11). The high-quality clean sequences with a 97% similarity cut-off were merged into the same operational taxonomic units (OTUs) [34]. The bioinformatics data were examined via the free online platform at https://www.omicstudio.cn/index (accessed on 5 October 2022). The QIIME (version 1.9.1) was constructed to analyze the α-diversity and the principal coordinate analysis (PCoA) with weight-Unifrac distance metric was constructed to calculate the β-diversity. The more stringent linear discriminant analysis (LDA) and effect size (LEfSe) was detected to further assesses bacterial taxa, as proposed by Segata et al. [35]. Spearman correlation heatmap was constructed by R [36] to show the correlations between bacterial community and fermentation parameters. The microbiota functional pathways were inferred using PICRUSt2 based on the information from the Kyoto Encyclopedia of Genes and Genomes (KEGG) database [37]. Bar graphs were presented and plotted using the software GraphPad Prism 9 (San Diego, CA, USA).

### 2.8. Statistical Analysis

Data on chemical composition, fermentation quality, and microbial characteristics of fresh and ensiled *Stipa grandis* were analyzed using a one-way analysis procedure of the SAS ver. 9.2 according to the statistical model: Y = μ + α + ε, where Y = observation, μ = overall mean, α = additive effect, and ε = error. Duncan’s tests separated significant differences, and *p* ˂ 0.05 was taken as statistical significance. The data values of the experiment are represented as the mean of the repeat among different treatments and the standard error of the mean.

## 3. Results

### 3.1. Lactic Acid Bacteria Strain Characteristics

The isolated and identified LAB strains in this study were Gram-positive, gas-for-glucose-negative, homofermentative, and catalase-negative. The strain ZX301 was cocci-shaped, while strain YX34 was rod-shaped bacteria. Two strains could normally grow at 10 and 30 °C, pH 3.5−8.0, and did not grow at temperatures of 5 °C or 50 °C or pH 3, while strain ZX301 could weakly grow at 45 °C. All isolated strains could grow normally under NaCl with concentrations of 3.0 and 6.5% (Table 1).

All strains could ferment arabinose, glucose, fructose, and cellobiose (Table 2). In addition, strain ZX301 could ferment xylose, and weakly ferment starch, while strain YX34 could not.

The 16S rDNA sequences were used to analyze molecular homological for observing divergences. The strain YX34 expressed a higher similarity with the *Lactiplantibacillus plantarum* according to the new classification methods of the Lactobacillus genus, and strain ZX301 described a higher similarity with *Pediococcus pentosaceus* (Table 3). They all supported the 100% similarities compared to their 16S rRNA gene sequence (Appendix A).

### 3.2. Silage Characteristics of Fresh Stipa Grandis

The constituent of DM and WSC was 50.71% and 2.52%, respectively. The CP, NDF, and ADF contents were 8.91, 75.46, and 39.79 on a DM basis, respectively. The amount of LAB, aerobic, coliform bacteria, and yeasts in the *Stipa grandis* was 5.56, 8.03, 6.25, and 7.01 log cfu/g FM, respectively. Fresh *Stipa grandis* molds were below the detectable level (Table 4).

### 3.3. Effect of Lactic Acid Bacteria Inoculant

The additive treatments extremely (*p* < 0.05) altered the WSC content, but had no significant (*p* > 0.05) effects on DM, CP, NDF, and ADF contents. The WSC content was remarkably (*p* < 0.05) decreased in the control silage relative to other treatments. Compared with the control silage, the pH values were extremely (*p* < 0.05) lower in ZX301, YX34, and P treatments, whereas the content of lactic acid was far (*p* < 0.05) higher in the inoculated silages. In addition, the contents of acetic acid in ZX301 and YX34 treatments were remarkably higher (*p* < 0.05) than that in the control silage, but remarkably lower (*p* < 0.05) than in the P treatment. The propionic acid and butyric acid contents were lower than the detectable levels in all treatments. The constituent of NH_3_-N was remarkably (*p* < 0.05) increased in the CK treatment relative to other treatments. The number of LAB was remarkably (*p* < 0.05) decreased in the YX34 and P treatments relative to the CK treatment. In contrast, no differences were measured in the number of aerobic bacteria among all treatments. The coliform bacteria, yeasts, and molds in all treatments were less than the detection level of <2.0 log_10_ cfu/g of FM (Table 5).

### 3.4. Microbial Diversity of Fresh Materials and Stipa Grandis after Ensiling

The Shannon and Simpson indices of the ZX301 treatment were extremely (*p* < 0.05) high relative to other treatments, whereas no differences were detected in ACE and Chao1 indices in all treatments. The Goods’ coverage in all treatments was about 100, which illustrated the accuracy and reproducibility of the sample assay, reflecting the actual situation of the bacteria in the sample (Table 6).

Overall, 857,373 raw reads were collected. Based on a 97% sequence identity threshold, a total of 801 OTUs were identified in fresh, control silage, and inoculated silage treatments. Of these, 66 OTUs were shared in all treatments, while 8, 5, 3, 2, and 2 OTUs were specific to the fresh, control, ZX301, YX34, and P treatments, respectively (Figure 1A).

For β-diversity, the PCoA score plot based on the weighted UniFrac distance metric indicated that the individuals in FM, CK, and ZX301 treatments could be markedly separated from each other. In addition, the individuals in YX34 and P treatments were also clearly separated from other treatments. However, no evident difference between YX34 and P was observed (Figure 1B).

At the phylum level, *Proteobacteria* (96.01%) was the most abundant phylum in fresh *Stipa grandis*, followed by *Actinobacteriota* (2.05%) and *Firmicutes* (1.86%). After 60 days of ensiling, the dominant phyla in CK, ZX301, YX34, and P treatments were *Firmicutes* (87.39% for CK, 88.50 for ZX301, 84.13 for YX34, and 96.14% for P), followed by *Proteobacteria* (11.37% for CK, 9.85 for ZX301, 15.29 for YX34, and 2.12% for P) (Figure 2A). Compared to FM treatment, the abundance of *Firmicutes* and *Proteobacteria* was significantly (*p* < 0.05) increased and decreased in CK, ZX301, YX34, and P treatments, whereas no significant differences were observed among CK, ZX301, YX34, and P treatments (Figure 2B).

At the genus level, the top three abundant genera in fresh *Stipa grandis* were *Pantoea* (94.16%), *unclassified_Microbacteriaceae* (1.80%), and *Lactiplantibacillus* (0.76%). After 60 days of ensiling, the *Lactiplantibacillus* (45.01%, 50.86%, 83.40%, and 43.06%, respectively) was the most dominant genera in CK, ZX301, YX34, and P treatments, while the subdominant genera in CK and P treatments were *Pantoea* (33.08% and 42.76%, respectively), in ZX301 it was *Pediococcus* (36.94%), and in YX34 it was *Enterobacter* (12.39%) (Figure 2C). Compared to FM treatment, the abundance of *Lactiplantibacillus* was significantly *(p* < 0.05) higher in CK, YX34, and P treatments, and the abundance of *Pantoea* was significantly (*p* < 0.05) lower in CK, ZX301, YX34, and P treatments. The abundance of *Pediococcus* in the ZX301 treatment was significantly (*p* < 0.05) higher than that in CK, YX34, and P treatments, whereas no significant differences were observed among CK, YX34, and P treatments. In addition, the YX34 treatment significantly (*p* < 0.05) decreased the abundance of *unclassified Enterobacteriaceae* than that in other treatments (Figure 2D).

The LEfSe analysis was used to reflect the variations of bacteria that most likely explained the differences in bacterial community structures of fresh *Stipa grandis* and after *Stipa grandis* silage (LDA score > 4.0). As shown in Figure 3, the results revealed that the fresh *Stipa grandis* before ensiling had various bacteria, but the types of bacteria were decreased after ensiling, with the FM treatment exhibiting a high abundance of *unclassified_Nitriliruptoraceae*, CK treatment exhibiting a high abundance of *unclassified_Enterobacteriaceae*, ZX301 treatment exhibiting a high abundance of *Pediococcus*, and P treatment exhibiting a high abundance of *Lactiplantibacillus*, at the genus level, respectively.

### 3.5. Relationships between Chemical Compositions, Fermentation Parameters, and Bacterial Community

The results show that the WSC content was positively correlated with *Hafnia_Obesumbacterium* and *Lactiplantibacillus*, but negatively correlated with *Pediococcus* and *unclassified_Enterobacteriaceae*. The DM and ADF contents were negatively correlated with *Escherichia_Shigella* and *Pediococcus*, respectively. The pH value was positively associated with *Enterobacter*. The NH_3_-N was positively associated with *Lentilactobacillus* and *unclassified_Enterobacteriaceae*, but negatively correlated with *Hafnia_Obesumbacterium*. AA content was positively correlated with *unclassified_Enterobacteriaceae*. In addition, the LA content was positively associated with *Hafnia_Obesumbacterium* and negatively associated with *unclassified_Enterobacteriaceae* (Figure 4).

### 3.6. Relationships between Chemical Compositions, Fermentation Parameters, and Bacterial Community

Notably, the main predicted functional genes at level 1 in the five treatments were assigned to metabolism (73.09−77.37%), genetic information processing (7.90−9.83%), and environmental information processing (5.30−8.09%), respectively (Figure 5A). Among these, the relative abundance of global and overview maps (41.55%), carbohydrate metabolism (12.87%), amino acid metabolism (6.65%), and membrane transport (6.03%) accounted for more than 5% of the enriched pathways among the five treatments (Figure 5B). At the three levels (Figure 5C), the genus associated with ABC transporters, bacterial secretion system, two-component system, metabolic pathways, microbial metabolism in diverse environments, glyoxylate and dicarboxylate metabolism, oxidative phosphorylation, sulfur metabolism, methane metabolism, quorum sensing, biofilm formation-*Escherichia coli*, biofilm formation-*Pseudomonas aeruginosa*, bacterial chemotaxis, flagellar assembly, and cationic antimicrobial peptide resistance was remarkably (*p* < 0.05) enriched in the FM treatment. The genus associated with purine metabolism, pyrimidine metabolism, and ribosome was remarkably (*p* < 0.05) enriched in the ZX301 treatment.

## 4. Discussion

The strategy of isolating novel LAB candidates in fermented forage has been widely used [38]. However, it is challenging to differentiate the species according to the phenotyping procedures to assign isolates [39]. The 16S rRNA sequence analysis method has proven to be highly effective in identifying the genus and species [26]. In the present study, strains ZX301 and YX34 were identified by 16SrRNA gene sequencing, which was determined to be homofermentative, and belong to *Pediococcus pentosaceus* and *Lactiplantibacillus plantarum*, respectively. The result was consistent with the findings of Yang et al. [40], who reported that the LAB genera, including *Lactobacillus* and *Pediococcus,* were the most frequently isolated microorganisms and exhibited a phylogenetically coherent treatment that clearly distinguishes them from other bacteria. Notably, the strains ZX301 and YX34 could grow normally at pH 3.5, which demonstrates an ability to grow in low pH environments of the strains and agree with the findings that homofermentative strains expressed high tolerance to a low-pH environment [41]. Generally, the optimum growth and reproduction temperatures of LAB should be below 45 °C [42]. In the present study, all strains could grow normally at temperatures of 10 °C or 30 °C, whereas strain ZX301 could weakly grow at 45 °C, which might be justified by the long-term evolution and natural selection on the unique environment of typical steppe [14]. In addition, all strains could ferment arabinose, glucose, fructose, and cellobiose, which suggests that the isolated strains in this study could widely use various fermentation substrates. Therefore, the unique traits of the two selected strains offer a warrant for using these LAB in silage making.

The material characteristics of the forage could directly affect the silage quality. In the present study, the DM content of *Stipa grandis* was 50.71%, consistent with the previously published [43]. Generally, WSC content is critical for successful silage. The WSC content found in this study was slightly lower than the level of 5% suggested by Amer et al. [44]. This study had the opposite result of Chiy et al. [45], who reported that Graminaceous species generally contain higher WSC content, likely as a result of prolonged drought stress in typical grassland, which could increase fructan hydrolysis and reduce the WSC content. However, the result of this study agrees with Hou et al. [8], who found the native grass in typical steppe had a low WSC content. In addition, the result of this study showed that the CP ingredient was lower, while NDF and ADF ingredients were higher in *Stipa grandis*, which agrees with Bu et al. [46].

Generally, successful silage depends mainly on the amount and type of microorganisms attached to the FM [47]. The previous study suggested that the optimum lactic acid bacteria content in raw materials should exceed 10^5^ cfu/g [48]. However, a lower content of LAB and higher numbers of undesirable microorganisms were found in FM in this study, indicating that direct ensilement would not be likely to succeed. Consequently, it is necessary to inhibit the growth of harmful bacteria during the early stages by using LAB additives for *Stipa grandis* silage.

As expected, the WSC, ADF, and NDF content was reduced compared to the fresh *Stipa grandis* after ensiling, mainly due to the conversion of available substrates to lactic acid by microbes related to the utilization of carbohydrates under anaerobic conditions [49]. Additionally, the acid hydrolysis of the digestible cell wall fraction might decrease the ADF and NDF content [50]. Notably, the inoculated silage treatments provided a benefit in preserving the WSC content relative to the CK treatment, which was in agreement with the research published by Sifeeldein et al. [17], who found that the LAB inoculants could inhibit the loss of WSC fermented by speeding up the process of LA fermentation to reduce pH value rapidly. However, a stable DM content was observed between CK and inoculated silage treatments, which agrees with the report that a combination of *L. acidophilus* and *L. plantarum* leads to a comparable DM content with controls [51]. The result might be because lactic acid fermentation predominantly occurred in all treatments, as is shown in Table 5, which will result in minimal DM losses [52]. Compared to the fresh material of *Stipa grandis*, a reduction in the CP content was detected after ensiling, which might be due to the action of plant and microbial enzymes contributing to the conversion of plant protein to non-protein nitrogen (NPN). In addition, the absence of WSC in *Stipa grandis* limited the production of lactic acid and the decrease in silage pH, and then failed to inhibit the activity of proteinase [53]. Whereas, the CP content between CK and inoculated silage groups was stable, which is consistent with a previous study published by Muck et al. [54], who observed that bacterial inoculants provided only a small benefit in inhibiting proteolysis. According to a previous report, protease exhibits optimal activity generally at the condition of a silage pH > 4.5 [55]. This result might be due to the lower terminal pH (4.00–4.15) in silage of all treatments, which could limit the activity of proteolytic enzymes [39].

The LA content, pH value, and NH_3_-N level of silages could directly reflect the silage quality [56]. As expected, the inoculants of LAB in *Stipa grandis* silage increased the LA content and led to a lower pH value and NH_3_-N content compared to the CK treatment in this study, all of which were within the recommended range provided by [57]. This result might be due to the LAB stimulating the LA fermentation by utilizing soluble carbohydrates and then producing a large amount of lactic acid, which could continuously reduce the pH value, and then inhibit the activity of proteases from plants and microbes to reduce proteolysis [58,59,60]. Similarly, Sucu et al. [61] proved that the addition of lactic acid bacteria could rapidly reduce the pH value and NH_3_-N content by increasing the LA content.

The Goods’ coverage of all samples exceeded 99%, indicating that the depth of sequencing was adequately captured for bacterial community analysis [62]. Generally, forage after ensiling has low alpha diversity caused by the dominant lactobacillus, which can inhibit the growth of other microbes and reduce the diversity of the bacterial community [63]. However, in the present study, the Shannon and Simpson indices were higher in the ZX301 treatment than that in FM and other inoculated silage treatments, which might be explained by the antagonistic activity between the dominant genera sourced from exogenous LAB and other bacteria, which weaken the advantage of the dominant genera (Figure 2C), and then lead to higher α-diversity indices in ZX301 treatment [64,65].

The PCoA analysis based on the Weighted UniFrac distances was performed to visualize the similarity and distinction of bacterial communities among treatments. FM treatment was clearly separated from silages, indicating that ensiling influenced the bacterial community. This finding was in agreement with Zhao et al. [66], who reported that significant differences in the bacterial community composition were detected before and after ensiling. In addition, the treatments of YX34 and ZX301 were also well separated from the CK treatment, which suggests that the addition of LAB reconstructs the microbial community.

At the phylum level, the bacterial community in *Stipa grandis* silage involves a shift from *Proteobacteria* to *Firmicutes*. As is shown in Figure 2A. *Proteobacteria* was the most abundant phyla in the fresh *Stipa grandis*. After 60 days of fermentation, the abundance of the *Proteobacteria* phylum was obviously decreased, whereas the *Firmicutes* phylum increased to become the new dominant phylum, which can be commonly found in previous studies [67,68]. *Proteobacteria* can result in proteolysis by competing with LAB to utilize WSC and can be animal pathogens [69]. *Firmicutes* plays a pivotal role in producing LA and AA after the later stage in ensiling [70]. This study showed a similarity in the richness and composition of bacteria in the CK, ZX301, YX 34, and P treatments at the phylum level. The relative stability of the main bacteria might reflect the existence of the core microbiome [71].

At the genus level, the main microorganism in fresh *Stipa grandis* was *Pantoea*. Their portions decreased significantly and were converted to *Lactiplantibacillus plantarum* after fermentation, which is in agreement with the results of You et al. [72], who reported that *Lactobacillus* can inhibit the undesirable bacteria abundance of fresh material and become the dominant genus in *Caragana* silage at the later stage of ensiling. *Pantoea* is harmful to fermentation by weakening the reduction in the pH value [73]. *Lactobacillus* was the dominant genus in high-quality silage, which can flourish under acid environmental conditions and accumulate LA and reduce the pH value during the initiation of fermentation [10,11]. Therefore, the high sensitivity to the changes in pH of *Pantoea* and *Lactiplantibacillus plantarum* might be the main reason for the conversion of the dominant genus between fresh and silage of *Stipa grandis* [74]. In addition, compared to the CK treatment, the addition of ZX301 and YX34 increased the abundance of *Pediocossus* and *Lactiplantibacillus*, respectively, indicating that inoculated strains in this study were not outcompeted by the indigenous microbial population, which might be attributed to inoculants of the strains dominating and controlling the microbial events during silage fermentation [75]. However, compared to the ZX301 treatment, the abundance of *Enterobacter* increased in the YX34 treatment, which can compete for available carbohydrates with LAB and, finally, lead to a negative impact on the silage [76], indicating that the ZX301 was more favorable for inhibiting the harmful bacteria and improving the quality of *Stipa grandis* silage.

Silage fermentation is actually a process caused by microorganisms. The relationship between the chemical composition, fermentation parameters, and microbial community was elucidated using the correlation analyses in this research. In this study, a significantly higher abundance of *Lactiplantibacillus* was observed in the YX34 treatment, which positively correlated with the WSC contents. The result agreed with Xu et al. [77], who found that *Lactiplantibacillus* was positively related to the WSC and NH_3_-N contents. These results indicate that *Lactiplantibacillus* exhibited a positive effect on nutrient preservation [77]. In addition, the abundance of *Pediococcus* was increased in the ZX301 treatment, and inversely correlated with WSC content, which suggests that this genus was primarily responsible for the LA generation and sugar consumption during ensiling in the ZX301 treatment [66]. Interestingly, the result showed that *Enterobacter* was positively correlated with pH value, and the *unclassified_Enterobacteriaceae* was inversely correlated with WSC and LA content, while positively correlated with AA and NH_3_-N content. These findings are corroborated by Fang et al. [78], who reported that the genus of *Enterococcus* positively correlated with AA and negatively with LA. The results of this research might be due to the acid sensitivity of *Enterococcus* [79]. Furthermore, a previous study has shown that the genera *Enterobacter* and *unclassified Enterobacteriaceae* belong to the *Enterobacteriaceae* family, which has proteolytic activities and might compete for nutrients with LAB and form undesirable silage [80,81]. In the present study, the ZX301 and YX34 treatments reduced the abundance of *Enterobacter* and *unclassified Enterobacteriaceae*, respectively, and enabled them to become new inoculants in *Stipa grandis* silage. Unfortunately, *Hafnia_Obesumbacterium* genes were detected during the whole process of silage, and were positively correlated with WSC and LA content; although, their abundance was very low. The genes of *Hafnia_Obesumbacterium* belong to *Enterobacteriaceae* and could survive in a strict anaerobic environment and contribute to proteolysis during ensiling [82,83]. Taken together, the presence of *Hafnia_Obesumbacterium* at least partially explains the reduction in CP content after ensiling [84].

The 16S rRNA gene sequences of the *Stipa grandis* before and after ensiled were used to infer the KEGG function profiles of microbial communities based on PICRUSt2. The result of the present study showed that the predicted functions from KEGG level 1 to level 3 were generally consistent in all treatments. The KEGG level 2 in silage fermentation mainly involves six metabolism pathways, including global and overview maps, carbohydrate metabolism, amino acid metabolism, cofactors and vitamins metabolism, energy metabolism, and nucleotide metabolism, which could be commonly found in previous studies [84,85]. The expression of the carbohydrate metabolism pathway was increased after ensiled, which might be explained by the process of silage fermenting the available carbohydrates to short-chain fatty acids [66]. Similarly, Su et al. [86] found that the abundance of carbohydrate metabolism caused an increase in fermentation progress. The amino acid metabolism was decreased more in the ZX301 treatment than that in other treatments, which might be due to the higher abundance of *Pediococcus* in the ZX301 treatment, which can rapidly reduce the pH value during the early stage of silage, and then inhibit the amino acid metabolism of silage [62,87]. This finding was consistent with the previously published study by Bai et al. [88], who reported that the addition of LAB can convert amino acid metabolism by modulating the bacterial communities. Additionally, the process of silage increased nucleotide metabolism in the present study. These metabolism functions are associated with the synthesis of DNA and the regulation of cellular processes [89]. The result was in agreement with Li et al. [90], who reported that the abundance of nucleotide metabolism increased with fermentation. However, our results could not represent the actual function of silage bacteria only based on the predicted microgenomics. Further detailed microgenomics analyses are needed to explore the potential mechanism and reveal gene functions of bacteria in silage.

## 5. Conclusions

This study evaluated the effect of LAB strains screened from *Stipa grandis* before and after silage on the silage quality and the bacterial community of *Stipa grandis*. The results indicated that the addition of LAB can improve the ensiling performance of *Stipa grandis* silage by regulating the microbiota. The ZX301 strains isolated from the *Stipa grandis* silage can increase the relative abundance of *Pediococcus,* reducing the pH value and rapidly inhibiting undesirable microorganisms, and therefore express the potential possibility for improving silage fermentation in *Stipa grandis* silage. Overall, the strains of ZX301 can be used as silage additives in making high-quality *Stipa grandis* silage.

## Figures and Tables

**Figure 1 microorganisms-10-02463-f001:**
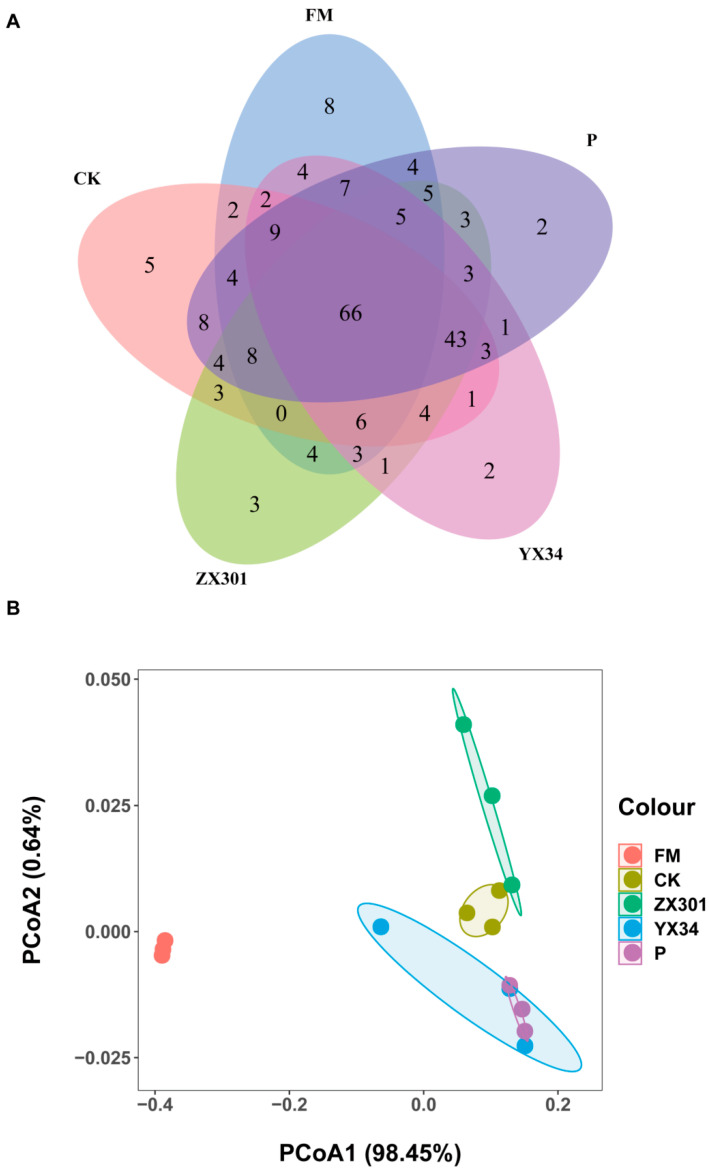
Microbial community of FM and *Stipa grandis* on 60 days of ensiling (*n* = 3): (**A**) Venn diagram representing the common and unique operational taxonomic units (OTUs) of FM and *Stipa grandis* on 60 days of ensiling. (**B**) Principal coordinates analysis (PCoA) of samples conducted based on weighted UniFrac distance. FM, fresh matter; CK, control group; ZX301, *pediococcus pentosaceus*-treated group; YX34, *lactobacillus planturum*-treated group; P, a commercial inoculant-treated group containing *lactobacillus plantarum*.

**Figure 2 microorganisms-10-02463-f002:**
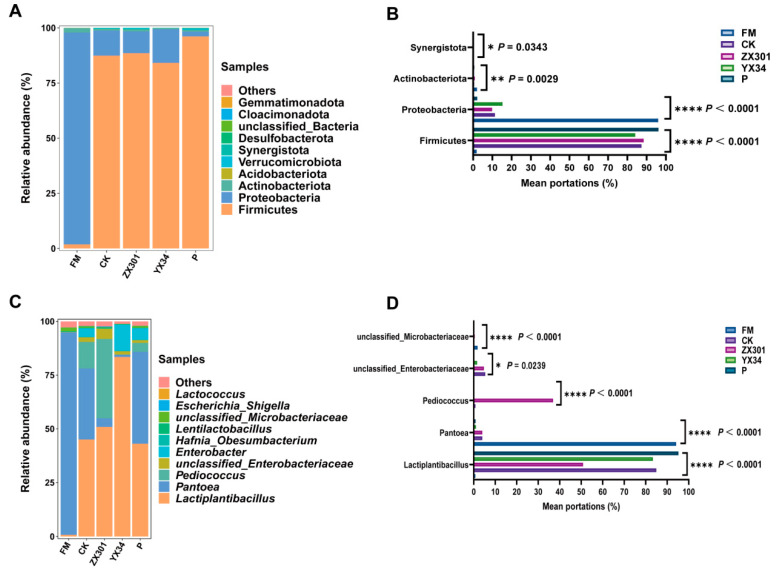
The bacterial community of FM and *Stipa grandis* on 60 days of ensiling (*n* = 3): (**A**) The bacterial community was shown at the phylum level. (**B**) The extended error bar plot displaying the significant differences among groups at the phylum level (**C**). The bacterial community was shown at the genus level. (**D**) The extended error bar plot displaying significant differences among groups at the genus level. Levels of significance are shown as follows: * *p* < 0.05; ** *p* < 0.01; **** *p* < 0.0001. FM, fresh matter; CK, control group; ZX301, *pediococcus pentosaceus*-treated group; YX34, *lactobacillus planturum*-treated group; P, a commercial inoculant-treated group containing *lactobacillus plantarum*.

**Figure 3 microorganisms-10-02463-f003:**
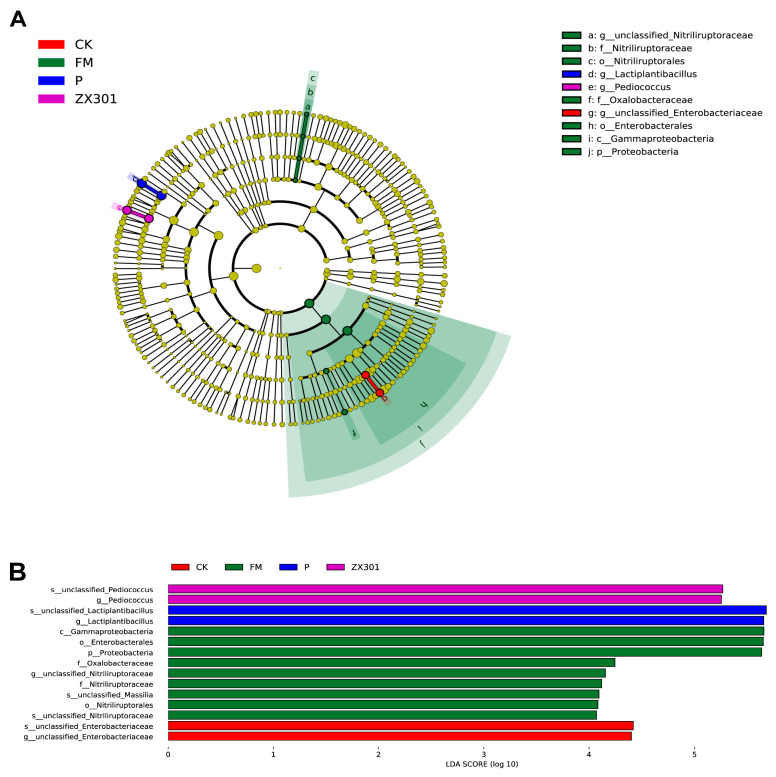
Linear discrimination analysis (LDA) coupled with effect size (LEfSe) analysis of the microbial community of the FM and *Stipa grandis* on 60 days of ensiling (*n* = 3): (**A**) Cladogram showing microbial species with significant differences between the FM and *Stipa grandis* on 60 days of ensiling. Red, green, purple, and blue represent different groups. Species classification at the phylum, class, order, family, and genus level are displayed from inner to outer layers. The red, green, purple, and blue nodes represent microbial species in the phylogenetic tree that play important roles in the CK, FM, ZX301, and P groups, respectively. Yellow nodes represent no significant difference between species. (**B**) Significantly different species with an LDA score greater than the estimated value (default score = 4). The length of the histogram represents the LDA score of different species in the FM and *Stipa grandis* on 60 days of ensiling. FM, fresh matter; CK, control group; ZX301, *pediococcus pentosaceus*-treated group; YX34, *lactobacillus planturum*-treated group; P, a commercial inoculant-treated group containing *lactobacillus plantarum*.

**Figure 4 microorganisms-10-02463-f004:**
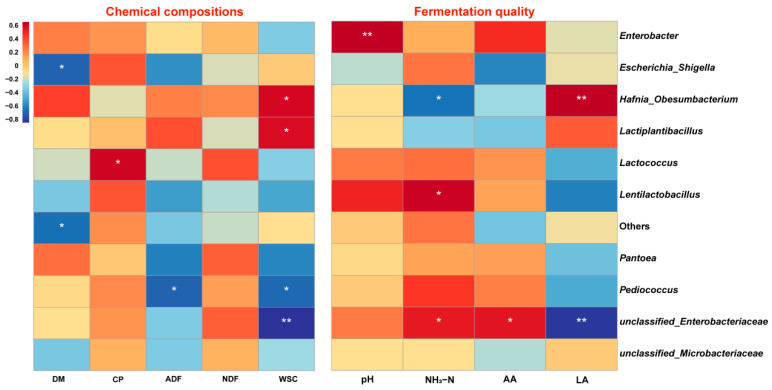
Heatmaps of Pearson’s correlations between dominant genera and chemical compositions and fermentation quality. Red represents a positive correlation, while blue represents a negative correlation. Levels of significance are shown as follows: * *p* < 0.05; ** *p* < 0.01.

**Figure 5 microorganisms-10-02463-f005:**
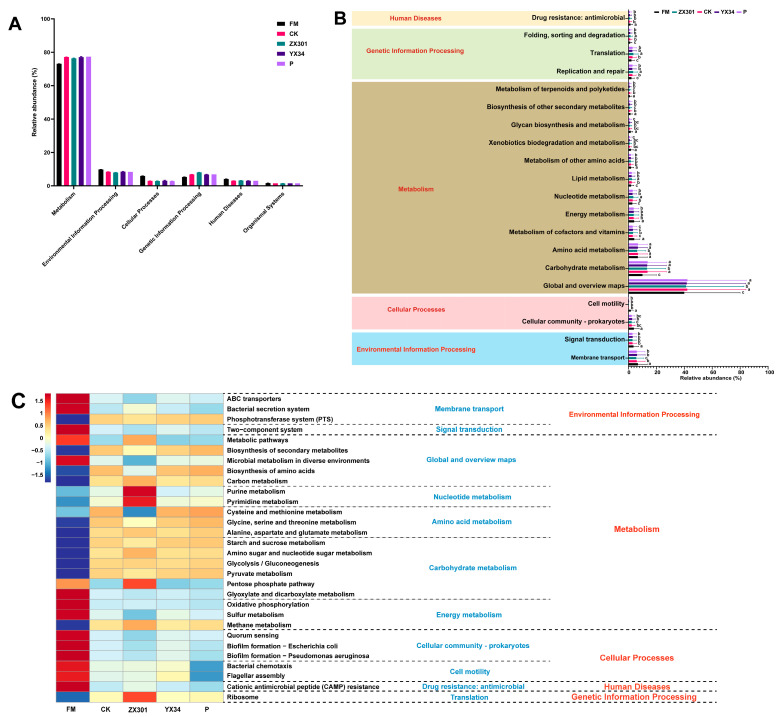
Dynamics of bacterial functional profiles analyzed by PICRUSt2 (*n* = 3): (**A**) Level 1 metabolic pathways. (**B**) Level 2 Kyoto Encyclopedia of Genes and Genomes (KEGG) ortholog functional predictions of the relative abundances of the top 20 metabolic functions. (**C**) Level 3 KEGG ortholog functional predictions of the relative abundances of the top 30 metabolic functions. FM, fresh matter; CK, control group; ZX301, *pediococcus pentosaceus*-treated group; YX34, *lactobacillus planturum*-treated group; P, a commercial inoculant-treated group containing *lactobacillus plantarum*.

**Table 1 microorganisms-10-02463-t001:** The selection of isolated lactic acid bacteria on the base of the morphological and physiological tests.

Characteristics	ZX301	YX34
Sources	*Stipa grandis* silage	*Stipa grandis* silage
Shape	Cocci	Rod
Gram stain	+	+
Gas for glucose	-	-
Catalase	-	-
Fermentation type	Homo	Homo
Growth at temperature		
5	-	-
10	+	+
15	+	+
30	+	+
45	w	-
50	-	-
Growth at pH		
3	-	-
3.5	+	+
4	+	+
5	+	+
6	+	+
7	+	+
8	+	+
Growth in NaCl (%)		
3	+	+
6.5	+	+

+, positive; -, negative; w, weakly positive. Homo, homofermentative.

**Table 2 microorganisms-10-02463-t002:** The carbohydrate fermentation characteristics of isolated lactic acid bacteria strains.

Items	ZX301	YX34
L-arabinose	+	+
Ribose	+	+
D-xylose	+	-
D-Galactose	+	+
D-Glucose	+	+
D-Fructose	+	+
D-Mannose	+	+
L-Sorbose	-	w
Mannitol	+	-
Sorbitol	+	-
Methyl-α D-Mannopyranoside	+	-
Methyl-α D-Glucopyranoside	+	-
N-Acetyl Glucosamine	+	+
Laetrile	+	+
Arbutin	+	+
Aescin	+	+
Salicin	+	+
Cellobiose	+	+
Maltose	+	+
Lactose	+	+
Honey disaccharide	+	+
Sucrose	+	+
Trehalose	+	+
Trisaccharide	+	-
Raffinose	+	-
Starch	w	-
Xylitol	+	-
Gentiobiose	w	+
D-Arabinitol	w	-
Gluconate	w	-

+, positive; -, negative; w, weakly positive. All strains gave negative results for glycerol, erythritol, D-arabinose, L-xylose, adonol, Methyl-βD-Xylopyranoside, L-rhamnose, dulcitol, Inositol, Inulin, Liver sugar, D-sondiose, D-xylose, D-tagatose, D-fucose, L-fucose, L-arabitol, 2-Keto-Gluconate, and 5-Keto-Gluconate.

**Table 3 microorganisms-10-02463-t003:** The results of isolated lactic acid bacteria on the base of 16S rRNA gene sequences.

Strain	Accession Number	16S rRNA Gene Sequencing Data (Closest Relative)	Similarity (%)
ZX301	KX886792.1	*Pediococcus pentosaceus* DSM 20336 T	100
YX34	NR_115605.1	*Lactobacillus plantarum* JCM 1149	100

**Table 4 microorganisms-10-02463-t004:** Chemical and microbial compositions of substrates before ensiling.

Items	*Stipa grandis*
Dry matter (g/kg FW)	50.71 ± 0.22
Crude protein (g/kg DM)	8.91 ± 0.10
Neutral detergent fiber (g/kg DM)	75.46 ± 0.72
Acid detergent fiber (g/kg DM)	39.79 ± 0.78
Water-soluble carbohydrates (g/kg DM)	2.52 ± 0.09
LAB (log_10_ cfu/g FM)	5.56 ± 0.10
Aerobic bacteria (log_10_ cfu/g FM)	8.03 ± 0.45
Coliform bacteria (log_10_ cfu/g FM)	6.25 ± 0.27
Yeasts (log_10_ cfu/g FM)	7.01 ± 1.33
Mold (log_10_ cfu/g FM)	ND

DM, dry matter; FM, fresh matter; log, denary logarithm of the numbers; cfu, colony-forming units; LAB, lactic acid bacteria; ND, not detected.

**Table 5 microorganisms-10-02463-t005:** Effects of additives on chemical composition, fermentation quality, and microbial compositions of *Stipa grandis* silage after 60 days.

Items	CK	ZX301	YX34	P	SEM	*p*-Value
DM (%)	50.97	50.41	50.99	50.29	0.0024	0.7162
WSC (% DM)	1.27 b	1.55 a	1.76 a	1.83 a	0.0007	0.0001
CP (% DM)	7.99	7.92	7.93	7.96	0.0308	0.0785
ADF (% DM)	70.35	70.15	72.09	69.62	0.0042	0.2207
NDF (% DM)	38.67	38.97	39.17	38.03	0.0031	0.7144
pH	4.15 a	4.00 b	4.01 b	4.01 b	0.0191	0.0007
Lactic acid (g/kg)	7.42 b	10.96 a	11.14 a	11.23 a	0.0479	0.0001
Acetic acid (g/kg)	4.68 a	3.79 b	3.95 b	3.07 c	0.0172	0.0001
Propionic acid (g/kg)	ND	ND	ND	ND		
Butyric acid (g/kg)	ND	ND	ND	ND		
NH_3_-N	1.38 a	0.87 b	0.74 b	0.88 b	0.0768	0.0015
LAB (log_10_ cfu/g FM)	8.20 a	7.28 ab	6.36 b	5.69 b	0.3468	0.0396
Aerobic bacteria (log_10_ cfu/g FM)	4.13	4.21	4.06	3.59	0.1293	0.3999
Coliform bacteria (log_10_ cfu/g FM)	ND	ND	ND	ND		
Yeasts (log_10_ cfu/g FM)	ND	ND	ND	ND		
Mold (log_10_ cfu/g FM)	ND	ND	ND	ND		

CK, control group; ZX301, *pediococcus pentosaceus*-treated group; YX34, *lactobacillus planturum*-treated group; P, a commercial inoculant-treated group containing *lactobacillus plantarum*; DM, dry matter; WSC, water-soluble carbohydrate; CP, crude protein; ADF, acid detergent fiber; NDF, neutral detergent fiber; NH_3_-N, ammonia nitrogen; LAB, lactic acid bacteria; log, denary logarithm of the numbers; cfu, colony-forming units; FM, fresh matter; ND, not detected; SEM, standard error of mean. Values in the same row with different letters are significantly different (*p* < 0.05).

**Table 6 microorganisms-10-02463-t006:** Effects of additives on bacterial alpha diversity of *Stipa grandis* silage after 60 days.

Items	FM	CK	ZX301	YX34	P	SEM	*p*-Value
OTUs	85	103	105	99	121	4.8563	0.2799
ACE	144.15	116.38	132.13	131.77	135.80	5.4811	0.7038
Chao1	130.11	118.12	129.53	123.04	134.34	4.5630	0.8751
Simpson	0.1140 b	0.2717 b	0.5979 a	0.2277 b	0.0924 b	0.0536	0.0034
Shannon	0.5305 b	1.0725 b	1.7554 a	0.7079 b	0.4884 b	0.1374	0.0025
Goods’ coverage	99.92 b	99.96 a	99.91 b	99.95 a	99.95 a	0.0001	0.0173

FM, fresh matter; CK, control group; ZX301, *pediococcus pentosaceus*-treated group; YX34, *lactobacillus planturum*-treated group; P, a commercial inoculant-treated group containing *lactobacillus plantarum*; OTUs, operational taxonomic units; SEM, standard error of mean; Values in the same row with different letters are significantly different (*p* < 0.05).

## Data Availability

Sequencing data for 16S rRNA gene sequence were stored in NCBI with BioProject accession number PRJNA891459.

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
