# Peer review of "Effects of Isolated LAB on Chemical Composition, Fermentation Quality and Bacterial Community of Stipa grandis Silage"

_microorganisms, 2022, doi:10.3390/microorganisms10122463_

Round 1

Reviewer 1 Report

Dear Authors,
the work is well structured and has good data processing.
It is a bit too long to follow, but quite flowing

Reviewer 2 Report

The manuscript presents the chemical composition, fermentation quality and bacterial community of Stipa grandis silage inoculated with isolated LAB that from the Stipa grandis and naturally fermented silage. It is a topic of interest to the researchers in the related areas but needs some revisions. The detailed comments are as follows:

1. L62-L63: Please pay attention to writing rules in the whole manuscript, such as L22 and L111 1×105 and so on.

2. L71-L72, please add the detailed methods of screening LAB strains.

3. L111, confused about  with all LAB strains, is that mixing all strains or inoculating them separately? Please describe the methods clearly.

4. Please harmonize the unit of chemical composition and fermentation quality in the tables, for example, NDF (g/kg DM).

5. L246, please change the 1 to 100.

6. Please pay attention to grammar and tenses in the whole manuscript, such as L415 and L440, may would be changed to might and so on.

7. Confused about L468-L469, usually Lactobacillus is considered a critical indicator in high-quality silage instead of Lactococcus, please modified the sentence.

8. L478, it is better to deleted was because of the active voice.

Reviewer 3 Report

In the work, the identification of LAB from Stripa grandis and the impact of selected LABs on the quality of silage and the shape of the bacterial community in the ensiled material over a period of 60 days was undertaken. Due to the potential use of Stripa as a feed material in ruminants feeding, the research should be considered appropriate.

The layout and structure of the work does not raise any objections in terms of methods and content.

The individual chapters of the work were properly prepared.

Some editorial errors or the need to supplement some information should be indicated.

Line 74: information contradictory to that in line 103, year 2020 or 2021? The identification of Lactobacillus strains was not performed in the same plant material that was ensiled?

Line 111: it is worth specifying on what basis the amount of LAB used for silage was determined.

I may have missed it, but the methodology for determining the characteristics of carbohydrate fermentation by isolated strains of bacteria was not found (Table 2).

Lines 182, 207, 217, 243, 323, 340: why include the content of individual tables or figures in the text of the work, it is enough to refer to the appropriate table.

Line 366: why "Strains" in a capital letter.

The first sentence in the summary seems redundant, that was the purpose of the research.

These minor comments do not affect the overall evaluation of the work.

After minor revision, the paper can be published in the Microorganisms Journal.

Reviewer 4 Report

Dear Authors,

 The paper entitled „Effects of isolated LAB on chemical composition, fermentation quality and bacterial community of Stipa grandis silage” focused on assessment the effects of lactic acid bacteria strains from isolated from the Stipa grandis on the silage quality and bacterial community of Stipa grandis after 60 days of fermentation process. Authors isolated and identified two LAB strains. Next Stipa grandis samples were ensilaged with those strains and with commercial additive for 60 days. Authors evaluated fermentation quality, chemical composition, and bacterial community of silages. The obtained results are interesting from cognitive and practical point of view. Authors found that one of strains (ZX301) could be used as silage additives in making high-quality Stipa grandis silage.

The manuscript is good written and can be published in present form.

Round 2

Reviewer 2 Report

Accept in present form.